# Effectiveness of a Mobile Application for Postpartum Depression Self-Management: Evidence from a Randomised Controlled Trial in South Korea

**DOI:** 10.3390/healthcare10112185

**Published:** 2022-10-31

**Authors:** Ji-Min Seo, Su-Jeong Kim, Hyunjoo Na, Jin-Hee Kim, Hyejin Lee

**Affiliations:** 1College of Nursing, Pusan National University, Yangsan 50612, Korea; 2College of Nursing, The Catholic University of Korea, Seoul 06591, Korea; 3Hanvit Internal Medicine Clinic, Busan 47837, Korea; 4Department of Nursing, Dong-Eui University, Busan 47340, Korea

**Keywords:** postpartum depression, self-management, mobile application, cognitive behavioural therapy, randomised controlled trial, Korea

## Abstract

This study examined the effectiveness of the Happy Mother mobile app developed for self-management of postpartum depression, based on cognitive behavioural therapy. A randomized controlled trial, with a pre- and a post-test design, was conducted in South Korea. Effectiveness was analysed using repeated measures ANOVA and Wilcoxon Signed Rank Test. We confirmed that the experimental group performed significantly more health promoting behaviours than the control group (F = 5.15, *p* = 0.007). However, there was no significant difference in postpartum depression, knowledge of depression, maladaptive beliefs, social support, sleep quality, and stress-coping behaviours between the two groups. The experimental group’s mood score increased by 1.79 ± 2.51 points, resulting in significant differences before and after the intervention (Z = −2.81, *p* = 0.005). The quality of sleep score in the experimental group increased by 1.48 ± 1.70 points and was also significantly different after the intervention (Z = −3.23, *p* = 0.001). The activity practice rate of the experimental group significantly increased by 30.27 ± 29.27% after using the app (Z = −2.81, *p* = 0.005). We found the app to be effective in promoting mothers’ health behaviour and improving their depressive mood.

## 1. Introduction

Depression has become a prevalent disease and critical health problem globally [1]. Among women, depression often occurs within a year of childbirth [2]. Depression beginning within six weeks of childbirth is called postpartum depression [3]. Mothers with postpartum depression experience lethargy, changes in appetite and weight, sadness, and guilt. Persistence of these symptoms can affect mother-child interactions; children’s emotional, behavioural, and cognitive development [4]; and lead to marital discord, and even divorce [5]. The incidence of postpartum depression varies by country. However, the recent incidence rate in South Korea was among the highest at 42.5% (compared with 10% in major European and North American countries) in a review of 143 postpartum depression-related studies from 40 countries [6].

Countries such as the US and the UK have legislated screening and education for postpartum depression, actively supporting early detection and coping [7,8]. In South Korea, the system of providing education through regional public and mental health centres, or by public collaboration with medical institutions, has not been implemented successfully [9]. Even when mothers recognise their poor mental state and try to actively cope, there is little time to do so because they have to take care of their babies [10]. To promote better mental health in South Korea, a potential option is a self-management mobile app where mothers can measure their own levels of depression and obtain information on postpartum depression management.

A recent meta-analysis reported that mobile app interventions are effective in reducing symptoms of mental disorders such as depression and anxiety [11]. For postpartum mothers, who are often sole caretakers, the highly accessible and convenient smartphones could be potentially effective instruments of self-management. As of 2019, about 95% of South Korean mothers were in their 20s and 30s, and more than 99% of them used smartphones [12], making these devices an advantageous option for health promotion [13]. As cost-effective, easy-to-access, and low-intensity interventions, health apps help mothers figure out their mental health [11]. Although a recent meta-analysis showed that postpartum depression apps were no more effective than standard interventions [14], apps’ advantages of accessibility suggest that their value should be reconsidered. Thus, its effects should be reconfirmed through further study.

Various psychotherapies have been used in the treatment of postpartum depression, including interpersonal psychotherapy, cognitive behavioural therapy (CBT), and dialectical behavioural therapy [15]. Several studies have reported that CBT has a significant therapeutic effect on postpartum depression [16,17,18,19].

To this end, the authors’ research team [20] developed a postpartum depression self-care mobile app, ‘Happy Mother’ (Appendix A), based on CBT principles. The superior quality of the app was previously confirmed through an app usability evaluation [20]. This study aimed to evaluate whether the use of the app, ‘Happy Mother,’ could reduce symptoms of postpartum depression among Korean mothers. In particular, Happy Mother was tested for its effects on postpartum depression, depression knowledge, maladaptive beliefs, social support, stress-coping behaviour, health-promoting behaviour, and sleep quality.

## 2. Materials and Methods

### 2.1. Study Design and Sample

We designed our study as a randomised controlled trial (RCT) (see Appendix A). Participants were recruited from 1 March to 13 November 2018, through Internet communities in three regions in South Korea. The Internet communities included mothers interested in sharing information, such as about housekeeping and childcare. Notices were posted on the relevant internet community boards, and mothers who wanted to participate replied. The respondents were screened for the following: (1) belonging to a risk or high-risk group, with a score of 9 or higher on the Edinburgh Postpartum Depression Scale [21,22]; (2) were within a year of childbirth; and (3) used Android-based smartphones. Those receiving psychiatric treatment were excluded.

We used the G-Power Analysis software (G-Power 3.1.9) to calculate the number of study participants required. In a prior study confirming the effectiveness of face-to-face postpartum depression interventions, the effect size (f) based on Cohen’s [23] criterion was 0.43 [24]. However, for this mobile app intervention, we applied a medium effect size. Based on repeated measures ANOVA, the minimum number of participants needed for each group was determined to be 28 for an effect size of 0.25, a statistical power of 0.80, and a significance level of 0.05, using two groups and three measurements.

The experimental and control group participants were determined by a pre-allocated random table. Allocation was concealed to avoid exposing the assignment order and information about the experimental and control groups. Each participant was met in person for the pre-test and to obtain written consent. They were informed about the collection of personal identification information required for post-testing, with information coded to protect participants’ confidentiality.

In total, 478 mothers responded that they would participate; after excluding 265 who did not meet the criteria and 113 who could not participate for personal reasons/commitments, random allocation was performed for the remaining 100; 50 participants were assigned to each group. The experimental group was provided with the postpartum depression self-management mobile app. A leaflet designed by the research team, comprising the definition, symptoms, causes, and coping methods of postpartum depression, was provided to the control group. After the experimental intervention, the control group could opt to use the app if they wanted.

During the eight-week intervention period, nine members of the experimental group were excluded for not using the smartphone app for more than one week and four other members for personal reasons; further, 14 members of the control group dropped out for personal reasons. Thus, the final sample included 73 participants (experimental group: 37; control group: 36) (see Appendix A). Eight participants were selected from the experimental group and interviewed personally to understanding their usage experience and suggestions for future app development.

### 2.2. The Happy Mother App

The mobile app, ‘Happy Mother,’ developed by the authors’ research team, provides various comprehensive services, such as information for depression management in mothers with postpartum depression, various self-management strategies, an algorithm for mood management, a bulletin board for communication with others, and phone contacts for suicide prevention counselling [20]. The results of the usability evaluation by the experts and mothers using the Mobile Application Rating Scale, an instrument that evaluates the general quality of the app, showed mean scores of 3.97 out of 5 points for experts and 3.64 out of 5 points for mothers [20].

The Happy Mother app’s framework consists of psychoeducation, managing mood and negative thoughts, increasing pleasant activity, and facilitating help-seeking behaviour [20]. The similarities between the app and basic principles of CBT are as follows. First, psychoeducation provides information on postpartum depression. This involves: self-diagnosing and understanding, treating, and overcoming postpartum depression; learning the role of a mother; and becoming a healthy mother. This is conceptualised as depression-related knowledge. Second, the management of mood begins with identifying the mother’s mood and sleep conditions. Low sleep quality adversely affects mood [25]; therefore, both mood and sleep are assessed. The app includes a diary where the management of mood and sleep are captured. This is conceptualised as postpartum depression and sleep quality. Third is the management of negative thoughts, which involves identifying and countering them; this includes thinking differently, having a daily motto, and writing in one’s diary. This is conceptualised as rectifying dysfunctional attitudes and stress-coping methods. Fourth is increasing pleasant activities, which enhances confidence through goal achievement, planning, and implementation; these activities are added to one’s Happiness Diary. This is conceptualised as health-promotion behaviour. Last, help-seeking behaviour facilitates coping with stress with the help of family members (e.g., spouse) and community resources; this includes obtaining a husband’s commitment to support his wife, awareness of relevant information, connecting to the helpline for telephone counselling, and posting on bulletin boards. Social support includes both the acts of providing and seeking help [26]. This can be provided by partners, family, peers, and colleagues, among others from the community and social media [27,28]. This is conceptualised as social support and facilitating help-seeking behaviour (the app menu details are shown in Appendix A).

### 2.3. Procedure

Five researchers were involved in the study, which ran from March 2018 to October 2019. The overall study was conducted as follows. The pre-test involved providing a structured questionnaire for the participant to complete. This took about 30 min and was conducted at participants’ homes or a place of their choosing. During the pre-test, Researcher B installed the Happy Mother app on the experimental group’s mobile phones and explained the basic functions. This group was then instructed to use the mobile app for eight weeks. There were no additional visits by researchers during this period. The number of times the app’s menu was accessed and the participant’s text input were sent to an administrator server (http://app2017.cafe24.com 6 July 2020). Researcher C checked the number of times the mobile apps were accessed by the experimental group and sent text messages to encourage weekly usage.

The post-testing for the groups was conducted in two rounds. The first round took place immediately after the usage period ended (T2: 8 weeks), three months after which, the second round took place (T3: 5 months). Structured questionnaires were used, and participants were allowed to choose to respond by e-mail or physical mail. The saved e-mail files were sent to Researcher D’s e-mail, and the mailed questionnaires were photographed and sent to Researcher D’s mobile phone. The post-tests took about 30 min to complete. The Researcher who conducted the pre and post-test did not know whether the participants who received the intervention were in the experimental or control groups.

### 2.4. Measures

The data were collected on factors such as age, educational level, religion, job, family income, marital satisfaction, number of pregnancies, planned pregnancy, childcare assistance, neonatal health problems, diagnosis of depression, diagnosis of mental illness except depression, and diagnosis of physical illness. The necessary permissions from developers and translators were obtained for all scales used.

#### 2.4.1. Postpartum Depression

Postpartum depression was measured using the Edinburgh Postnatal Depression Scale (EPDS), designed to assess what the participant felt over the past week. It comprises 10 questions [21,22] and the responses are measured on a 4-point Likert scale, with scores ranging from 0 to 30—the higher the score, the higher the level of postpartum depression. A score of 9 or higher was categorised as the risk group and a score of 13 or higher as high-risk [22]. Cronbach’s α for in this study was 0.82.

#### 2.4.2. Knowledge of Depression

Knowledge of depression was measured using the Parental Depression Literacy Scale developed by Griffiths, Christensen, Jorm, Evans, and Groves [29] and translated by Jeong et al. [30]. We selected 14 questions (out of the original 18) that corresponded to the information provided in the mobile app. These covered misperceptions of depression and treatment, depression-related knowledge, and treatment-related knowledge. Participants received 1 point for a correct answer and 0 for an incorrect answer. The scores ranged from 0 to 14—the higher the score, the higher the knowledge of depression.

#### 2.4.3. Maladaptive Beliefs

Maladaptive beliefs were measured using the Dysfunctional Attitude Scale-A, developed by Weissman and Beck [31] and translated by Kwon [32]. The original scale consists of 40 questions. We used 25 questions that Kwon [32] extracted through factor analysis, measured on a 7-point Likert scale, ranging from 1 (totally disagree) to 7 (totally agree). Scores ranged from 25 to 175 points—the higher the score, the higher the dysfunctional belief. Cronbach’s α for the scale was 0.91.

#### 2.4.4. Social Support

Social support was measured using a scale developed by Park [33]. The scale assesses an individual’s perception of the social behaviours of his family and neighbours, and comprises 25 questions measured on a 5-point Likert scale, ranging from 1 (not at all) to 5 (very much). The scores ranged from 25 to 125—the higher the score, the more social support the participants had. Cronbach’s α for the scale was 0.96.

#### 2.4.5. Stress-Coping Behaviour

Coping with stress was measured using the Ways of Coping Checklist developed by Folkman and Lazarus [34], modified by Jeon [35]. The original scale comprises 62 questions; we used 40 questions on four subscales derived by Jeon in his modification: 13 questions on problem-focused coping, 5 on seeking social support, 12 on tension reduction, and 10 on wishful thinking. Responses were recorded on a 4-point Likert scale, ranging from 0 (does not apply/or is not used) to 3 (used a great deal). The scores ranged from 0 to 120—the higher the score, the higher the dependence on any coping method. Cronbach’s α for this ranged between 0.40–0.80 for the four subscales.

#### 2.4.6. Health-Promoting Behaviour

Health-promoting behaviour was measured using the Health-Promoting Lifestyle Profile-Ⅱ scale by Walker, Sechrist, and Pender [36] and translated by Yun and Kim [37]. It comprises 52 questions across the following domains: physical activity, interpersonal relationship, and stress management. Responses were recorded on a 4-point Likert scale, ranging from 1 (not at all) to 4 (always). The scores ranged from 22 to 88—the higher the score, the higher the level of practice of the health-promoting behaviour. Cronbach’s α for the scale was 0.86.

#### 2.4.7. Sleep Quality

Sleep quality was measured on the Pittsburgh Sleep Quality Index, developed by Buysse, Reynolds, Monk, Berman, and Kupfer [38] and translated by Cho et al. [39]. It comprises 18 questions measuring subjective sleep quality, latency, duration, efficiency, disturbance, use of sleep medication, and daytime dysfunction. Each component was rated from 0 to 3 points, with total score ranging from 0 to 21―the higher the score, the worse the sleep quality. Cronbach’s α for this scale was 0.63.

#### 2.4.8. Self-Recorded Data in the Happy Mother App Diary

Mood, sleep quality, and activity were measured through the Happiness Diary, which captured participants’ self-recorded information. The mood scores ranged from 1 to 10―the higher the score, the better the current mood. Sleep quality ranged from 1 to 10, with high scores reflecting high sleep quality the previous night. The activity practice rate captured the percentage of activities actually executed compared to planned activities.

### 2.5. Experiences Using the App

Open-ended interviews were conducted with eight participants to explore the experiences of the mothers using the app in the experimental group. Researcher B recruited participants and obtained consent to participate in the study. The interviews were recorded and transcribed; each interview took about 30 min and was conducted at the participants’ house or where the participants felt comfortable. The main interview questions were:Please describe your experience using the app.Please describe how you thought the app was helpful.Please describe what needs to be improved or supplemented in the app.

### 2.6. Ethical Considerations

This study was approved by the University Institutional Ethics Committee. Consent was obtained from all participants and the need for a pre-test and personal mobile phone numbers was explained. All participants received $9 gift certificates.

### 2.7. Statistical Analysis

We analysed the data using IBM SPSS Statistics for Windows, version 25.0. Normality of distribution was assessed for all datasets with the Shapiro–Wilk test. Assessments of the homogeneity of variance between the experimental and control groups were conducted by χ^2^ test, Fisher’s exact test, and the Mann–Whitney U test. A parametric method was used when normality of distribution was satisfied, and a nonparametric method was used when it was not. The effectiveness verification used 2 × 3 repeated measures ANOVA to confirm the interaction of time and groups. The differences between the two groups’ pre-(T1) and post-first round (T2) variations, as well as pre-(T1) and post-second round (T3) variations were analysed using the independent t-test and Mann–Whitney U test. The score changes recorded by the experimental group in the Happiness Diary were analysed using the Wilcoxon Signed Rank Test. As there were only a few diary users in the post-second round measurement, only differences between the pre-measurement and the measured values (T2) at the end of the intervention period are noted.

Interview data were analysed using Braun and Clarke’s [40] thematic analysis. First, the process of performing open-coding and categorising was completed; then, the central themes were derived. To improve our qualitative analysis, credibility, auditability, fittingness, and conformability were verified [41].

## 3. Results

### 3.1. General Demographics and Baseline Comparisons

Table 1 shows the participants’ general demographics. There were no significant differences between the experimental and control groups regarding these variables; homogeneity was therefore confirmed (See Table 1 and Table 2).

### 3.2. Effects of the Intervention

From the repeated measures ANOVA analysis, when comparing the experimental and control group’s depression-related knowledge scores, we found no significant difference between groups, time points, or group–time interaction. In terms of dysfunctional attitude, social support, problem-focused coping, and sleep quality there were no significant differences between the groups, but there were significant differences at the time points. In terms of social support, tension reduction, and wishful thinking, there were no significant differences between groups, time points, or group–time interaction. However, for health-promoting behaviour, although there was no significant difference between groups, there was a significant difference in time points and in group–time interactions (See Table 3).

### 3.3. Experiences in Using the App

From our conversations with eight mothers in the experimental group who used the app, seven themes emerged (See Appendix A).

(1)How the app was helpful

(Theme 1) Self-monitoring and planning life using the Happiness Diary

The mothers reported being satisfied with capturing their moods and their sleep quality in the Happiness Diary. They stated that being able to check their condition and plan pleasant activities was helpful. They referenced the convenience of reviewing their mood, sleep quality, and activity rate through a graph.

(Theme 2) Functions to help manage depression

The mothers reported that various functions helped them identify their depression, change to more positive thinking, and share their thoughts with their husbands.

(Theme 3) Helpful parenting information

The mothers liked the educational aspects of the app, including information about the process of becoming a mother and parenting.

(Theme 4) Psychological comfort through the bulletin board

When mothers left messages on the bulletin board, our research team provided feedback. The goal was to listen to the mothers’ troubles, console them, and provide solutions.

(2)Improvements needed

(Theme 1) Supplement the functions of the Happiness Diary

The mothers said that certain functions needed to be clarified. Specifically, they said it was not easy to understand how sleep quality was expressed in numbers. Additionally, there were no alerts/reminders to encourage planned activities.

(Theme 2) More training on altering dysfunctional thoughts

The mothers said that they did not use the ‘Thinking Differently’ function frequently, and that they would need more practice on altering dysfunctional thinking.

(Theme 3) Periodic updates

The mothers noted that the information in the app was limited and should be updated periodically.

## 4. Discussion

This study attempted to determine whether the app developed by this research team was effective in alleviating the symptoms of mothers with postpartum depression. The app was found to be significantly effective in improving health-promotion behaviours of mothers, with the implication that long-term use could further improve depression outcomes.

We ran the RCT to determine whether using a self-management app could reduce postpartum depression in mothers in South Korea. Based on CBT principles, we tried to confirm the effectiveness of the app in increasing mothers’ knowledge of depression, changing maladaptive beliefs, improving social support, stress-coping behaviour, health-promoting behaviour, and sleep quality. For future app development, we also obtained suggestions for improvement through open-ended interviews with mothers using the app.

As a counselling approach, CBT is used to treat various psychological and mental illnesses. It applies cognitive strategies that change a person’s thinking patterns and behavioural strategies that help shift the person away from unhelpful or harmful behaviours [42]. Cognitive restructuring is an important approach to treat depression [43]. The Happy Mother app attempts to achieve cognitive restructuring through its ‘Thinking Differently’ function, which intends to help mothers apply depression management in real life by shifting negative thoughts into positive ones. Additionally, the app aims to help mothers set goals for their daily activities to boost and improve their moods. Setting goals provides motivation and maintains consistent behaviour [42]. Our trial attempted to identify whether mothers’ depression-management behaviour changed by using an app with these functions.

We found that using the app did not lead to any difference in postpartum depression between the experimental and control group after the eight-week intervention. This is similar to the meta-analysis results of seven studies applying depression-management mobile apps to mothers, where the effect of the mobile app intervention was no different from that of standard intervention [14]. However, a prior meta-analysis of RCTs examining the effects of depression-management apps found that smartphone apps had a depression-ameliorating effect when used by participants with moderate to mild depression [44]. However, the participants were the general population with depression symptoms, not mothers, and the sample size was large. Additionally, more than half of the trials had an app intervention application period of longer than eight weeks. In contrast, in the experimental group of our study, the baseline depression score was high, and the application period of the app intervention was short (only eight weeks). The implication is that the app may be more effective for mothers with mild or moderate depression scores with an application period of at least three to five months.

Furthermore, postpartum depression is often affected by a lack of communication in the family, an absence of childcare assistance, and the burden of childcare [15]. To compensate for the lack of communication, we created a bulletin board for communication among mothers and medical staff; however, the usage rate was low (Appendix A). While childcare information and related apps were introduced, this apparently did not reduce the burden of childcare felt by the mothers.

In contrast, the health-promoting behaviour of the experimental group increased significantly compared with that of the control group. We believe that the use of ‘activity’ in the app diary helped promote healthy behaviour in the experimental group. ‘Activity’ encompassed a wide range of actions, such as stretching, making appointments with friends, or even taking warm-water baths; participants could perform pleasant activities that they planned daily and check how many of the activities they actually performed each evening. The analysis of changes in activity rates showed that the average weekly rate rose from 51.2% to 81.5% during the eight-week intervention. Several studies have reported that as physical activity, social exchange, and relaxation times increase, sleep quality improves [45] and depression decreases [46]. The app intervention had a significant effect on encouraging health-promoting activities, with the implication that long-term use could further improve depression.

We found that knowledge of depression in the experimental group did not improve significantly compared with the control group. However, the pre-test scores of depression knowledge were high. The experimental group scored 12.11 and the control group 12.17 of 14 possible points. Therefore, the scope for knowledge improvement was limited. In a study by Jeon and Park [47], knowledge scores about diabetes before and after using a diabetes self-management app were compared with no significant difference. Similarly, in that study, the authors explained that as the knowledge pre-score was 78 out of 100, there were limitations to measuring knowledge increase.

The experimental group’s maladaptive beliefs did not differ statistically significantly from the control group. Based on cognitive reconstructing, the app’s ‘Thinking Differently’ function was meant to help change negative attitudes that cause depression [43]. This function allowed mothers to practice turning negative thoughts into positive ones. However, only 43.2% of the participants used this function more than once a week during the intervention period. In a study by Mantani et al. [48], depression was significantly reduced by applying a CBT-based app to patients diagnosed with depression over nine weeks. The app consisted of eight sessions, such as self-monitoring, behavioural activation, and cognitive reconstructing, with each session lasting one week. Cartoon characters were used to provide detailed descriptions of CBT principles and skills in the form of instant messenger. Participants faithfully performed the assignment for each session after listening to the explanations. As a result, 88% of the participants completed six out of eight sessions. Given the ‘Thinking Differently’ function’s low usage, the app may have lacked sufficient background on the concept and more training was needed for the same. In the future, the app should include a better explanation of the principles of cognitive reconstruction as well as improved training on it.

We found that the app was ineffective in stress coping for mothers. Specifically, problem-focused coping increased significantly over time, but there was no difference between the groups. Lazarus and Folkman [49] state that problem-focused coping to overcome stress means defining problems to create alternative solutions, weighing alternatives in terms of gain and burden, and acting on the most appropriate option. The mood-management algorithm was based in the mood area of the Happiness Diary; the algorithm started when the mother’s mood scores were at a certain level. At that certain score, an alternative to managing the mood was presented. Just presenting an alternative increases awareness of how to deal with one’s mood, which increases motivation and the possibility of implementing an alternative.

Social support did not differ significantly between the two groups. A prior study investigating 3310 postpartum mothers in cohorts found that the lower the social support, the more severe the postpartum depression [50]. Happy Mother included the functions, ‘be with your husband’ and ‘bulletin board’ to strengthen social support, but only nine (24.32%) and five (13.51%) participants, respectively, used these functions. The interview results confirmed that the bulletin board was not actively used. The team periodically checked bulletin board posts and provided feedback, but bulletin board use was difficult to encourage.

Sleep quality did not differ significantly between the experimental and control groups; although still low, both groups did improve over time. Information such as sleep hygiene was included in the app; however, as babies under the age of one wake up once or twice at night [51], there were limits to improving the mother’s sleep.

The diary was the function that the experimental group used the most, and the postpartum Q&A was used the least. As the mother’s health condition and the baby’s development phase were constantly changing, the app provided insufficient information. Additionally, the usage rate was low due to a lack of app information updates, as mentioned in the interviews.

When analysing our results comprehensively, we noted the following improvements needed to support postpartum depression management in the future. First, if more alarms or reminders that encourage mothers to plan pleasant activities are introduced, the activity practice rate will increase. The importance of altering dysfunctional thinking should be better explained, with training provided for this. The app should encourage active communication with experts and other mothers. Finally, mothers would use the app more frequently if it updated information regularly.

Our study has three notable strengths. First, the internal validity of our results was enhanced through the adoption of an RCT. We also reduced bias by blinding researchers who performed random allocations, collected data, and conducted statistical analysis. Second, we implemented an app with functionality based on CBT, an effective non-drug treatment for depression. Third, we interviewed mothers who participated in the intervention to explore app usage experience and possibilities for future improvement.

However, several limitations warrant comment. First, the average times the app was used by the experimental group was about three times a week, making it difficult to determine the app’s effect. We consistently sent encouraging text messages to those who accessed it less than once a week, but our ability to increase usage this way was limited. Second, the app was for Android phones only, which excluded iPhone users, thus impacting our random sampling. Third, this RCT was not prospectively registered.

## 5. Conclusions

We aimed to assess the usefulness of an app for mothers’ self-management of postpartum depression, depression knowledge, maladaptive beliefs, social support, stress-coping behaviour, health-promoting behaviour, and sleep quality. The CBT-based app’s effectiveness was evaluated through an RCT. Results confirmed that the app was effective in promoting healthy behaviour.

Ultimately, the concept of the app was to reduce postpartum depression by supporting mothers’ self-management behaviour. As helping patients manage themselves is a central nursing task, such an app could be used in self-management nursing intervention programmes for postpartum depression. Considering that the trial period for the app was short and the usage rate low, further research is needed to build on our results.

## Figures and Tables

**Table 1 healthcare-10-02185-t001:** Homogeneity Test for General Characteristics between Groups (N = 73).

Characteristics	Categories	Experimental Group (n = 37)	Control Group (n = 36)	χ^2^/t/z	*p*
		n(%)/M ± SD	n(%)/M ± SD		
Age (years)	<30	3 (8.1)	7 (19.5)	1.98	0.371
	30–34	20 (54.1)	17 (47.2)		
	≥35	14 (37.8)	12 (33.3)		
		33.54 ± 3.30	33.36 ± 4.47	0.20	0.845
Education level	High school	1 (2.7)	2 (5.6)	0.39	0.824
	University	31 (83.8)	29 (80.5)		
	≥Graduate school	5 (13.5)	5 (13.9)		
Religion	Christianity	8 (21.6)	5 (13.9)	1.32	0.724
	Catholic	2 (5.4)	2 (5.6)		
	Buddhism	10 (27.0)	8 (22.2)		
	No religion	17 (46.0)	21 (58.3)		
Job	Yes	13 (35.1)	9 (25.0)	0.89	0.345
	No	24 (64.9)	27 (75.0)		
Income (in US dollars)	<1800	1 (2.7)	1 (2.8)	1.47	0.833
	1800–2699	14 (37.8)	10 (27.8)		
	2700–3599	10 (27.0)	9 (25.0)		
	3600–4499	7 (18.9)	8 (22.2)		
	≥4500	5 (13.6)	8 (22.2)		
Marital satisfaction	Dissatisfaction	2 (5.4)	1 (2.8)	1.51	0.680
	Moderate	12 (32.4)	16 (44.4)		
	Satisfaction	18(48.7)	16 (44.4)		
	Very satisfied	5 (13.5)	3 (8.4)		
Number of pregnancies		1.97 ± 1.09	2.08 ± 1.57	−0.17	0.869 ^‡^
Planned pregnancy	Yes	18 (48.6)	21 (58.3)	0.69	0.407
	No	19 (51.4)	15 (41.7)		
Method of feeding	Breastfeeding	7 (18.9)	11 (30.6)	1.79	0.408
	Mixed breastfeeding	11 (29.7)	7 (19.4)		
	Bottle feeding	19 (51.4)	18 (50.0)		
Childcare assistance	Yes	33 (89.2)	33 (91.7)		1.000 ^†^
	No	4 (10.8)	3 (8.3)		
Neonatal health problem	Yes	5 (13.5)	3 (8.3)		0.711 ^†^
	No	32 (86.5)	33 (91.7)		
Diagnosis of depression	Yes	3 (8.1)	3 (8.3)		1.000 ^†^
	No	34 (91.9)	33 (91.7)		
Diagnosis of mental illness	Yes	0 (0.0)	1 (2.8)		0.493 ^†^
(other than depression)	No	37 (100.0)	35 (97.2)		
Diagnosis of physical illness	Yes	7 (18.9)	2 (5.6)		0.152 ^†^
	No	30 (81.1)	34 (94.4)		

^†^ Fisher’s exact test; ^‡^ Mann–Whitney U Test.

**Table 2 healthcare-10-02185-t002:** Homogeneity Test of Dependent Variables between Groups (N = 73).

Variables	Experimental Group (n = 37)	Control Group (n = 36)	t/z	*p*
	M ± SD	M ± SD		
Knowledge of depression	12.11 ± 1.31	12.17 ± 1.16	−0.02	0.982 ^†^
Maladaptive beliefs	93.84 ± 19.17	91.44 ± 18.27	0.55	0.587
Social support	85.95 ± 15.09	84.17 ± 14.44	0.52	0.608
Stress-coping behaviour Problem-focused coping	22.89 ± 4.82	23.17 ± 3.89	−0.07	0.947 ^†^
Seeking social support	9.35 ± 2.54	8.92 ± 2.93	−0.46	0.647 ^†^
Tension reduction	20.86 ± 3.15	21.36 ± 3.32	−0.66	0.515
Wishful thinking	21.30 ± 3.50	21.78 ± 3.16	−0.47	0.637 ^†^
Health-promoting behaviour	40.84 ± 7.52	39.89 ± 7.58	0.54	0.593
Sleep quality	9.73 ± 3.11	10.25 ± 3.60	−0.57	0.571 ^†^
Postpartum depression	13.95 ± 4.56	15.03 ± 5.35	−0.93	0.355

^†^ Mann–Whitney U Test.

**Table 3 healthcare-10-02185-t003:** App Effects on Dependent Variables (N = 73).

Variables	G	T1	T2	T3	S	F/Z	*p*	T2-T1 (M ± SD)		T3-T1 (M ± SD)	
		M ± SD						t/z	*p*	t/z	*p*
Postpartum depression	Exp. (n = 37)	13.95 ± 4.56	10.70 ± 4.64	9.84 ± 5.79	G	2.36	0.129	−3.24 ± 3.99		−4.11 ± 5.31	
	Cont. (n = 36)	15.03 ± 5.35	13.03 ± 6.19	11.47 ± 6.05	T	23.37	<0.001	−2.00 ± 5.61		−3.56 ± 4.46	
					G×T	0.59	0.555	−1.09	0.278	−0.48	0.632
Knowledge of depression	Exp. (n = 37)	12.11± 1.31	11.78 ± 1.25	12.08 ± 1.38	G	0.05	0.830	−0.32 ± 1.23		−0.03 ± 1.09	
	Cont. (n = 36)	12.17 ± 1.16	12.08 ± 1.02	11.89 ± 1.51	T	1.23	0.296	−0.08 ± 0.97		−0.28 ± 1.41	
					G×T	1.65	0.195	−0.74	0.457 ^†^	0.44	0.661 ^†^
Maladaptive beliefs	Exp. (n = 37)	93.84 ± 19.17	89.65 ± 21.17	87.19 ± 19.27	G	0.00	0.947	−4.19 ± 13.68		−6.65 ± 19.90	
	Cont. (n = 36)	91.44 ± 18.27	91.89 ± 21.01	88.19 ± 21.45	T	4.04	0.024	0.44 ± 12.29		−3.25 ± 13.94	
					G×T	0.93	0.388	−1.52	0.133	−0.84	0.402
Social support	Exp. (n = 37)	85.95 ± 15.09	88.54 ± 11.50	90.38 ± 14.91	G	0.58	0.448	2.59 ± 11.46		4.43 ± 14.85	
	Cont. (n = 36)	84.17 ± 14.44	85.31 ± 18.75	87.81 ± 18.61	T	4.27	0.016	1.14 ± 11.86		3.64 ± 9.92	
					G×T	0.14	0.870	0.53	0.596	0.27	0.789
Stress-coping behaviour											
Problem-focused coping	Exp. (n = 37)	22.89 ± 4.82	24.05 ± 3.18	24.14 ± 3.10	G	0.02	0.888	1.16 ± 3.14		1.24 ± 3.63	
	Cont. (n = 36)	23.17 ± 3.89	23.81 ± 2.67	23.81 ± 3.13	T	4.23	0.021	0.64 ± 3.33		0.64 ± 3.47	
					G×T	0.40	0.641	−0.93	0.351 ^†^	−0.88	0.378 ^†^
Seeking social support	Exp. (n = 37)	9.35 ± 2.54	9.84 ± 1.94	9.92 ± 2.06	G	0.87	0.353	0.49 ± 2.19		0.57 ± 2.27	
	Cont. (n = 36)	8.92 ± 2.93	9.39 ± 2.11	9.44 ± 2.62	T	3.01	0.060	0.47 ± 2.27		0.53 ± 2.35	
					G×T	0.00	0.993	−0.30	0.766 ^†^	−0.29	0.768 ^†^
Tension reduction	Exp. (n = 37)	20.86 ± 3.15	21.57 ± 2.90	21.41 ± 2.71	G	0.01	0.930	0.70 ± 3.13		0.54 ± 3.16	
	Cont. (n = 36)	21.36 ± 3.32	21.33 ± 3.60	21.31 ± 2.98	T	0.49	0.612	−0.03 ± 2.87		−0.06 ± 2.47	
					G×T	0.62	0.542	1.04	0.302	0.90	0.372
Wishful thinking	Exp. (n = 37)	21.30 ± 3.50	21.57 ± 3.24	21.68 ± 3.15	G	0.03	0.859	0.27 ± 3.06		0.38 ± 3.30	
	Cont. (n = 36)	21.78 ± 3.16	21.08 ± 3.23	21.36 ± 3.10	T	0.17	0.846	−0.69 ± 3.21		−0.42 ± 4.07	
					G×T	0.81	0.449	−1.13	0.259 ^†^	−0.55	0.583 ^†^
Health-promoting behaviour	Exp. (n = 37)	40.84 ± 7.52	44.11 ± 8.11	46.97 ± 9.72	G	2.69	0.105	3.27 ± 4.99		6.14 ± 8.27	
	Cont. (n = 36)	39.89 ± 7.58	42.50 ± 8.67	40.97 ± 9.40	T	10.08	<0.001	2.61 ± 7.63		1.08 ± 6.29	
					G×T	5.15	0.007	0.44	0.663	2.93	0.005
Sleep quality	Exp. (n = 37)	9.73 ± 3.11	8.76 ± 3.20	8.30 ± 3.37	G	0.80	0.373	−0.97 ± 2.86		−1.43 ± 3.51	
	Cont. (n = 36)	10.25 ± 3.60	9.67 ± 3.41	8.64 ± 3.46	T	8.38	0.001	−0.58 ± 3.56		−1.61 ± 3.64	
					G×T	0.31	0.715	−0.81	0.418 ^†^	−0.06	0.956 ^†^
Self-recorded data in Happiness Diary											
Mood	Exp. (n = 23)	4.19 ± 1.85	5.98 ± 1.90					1.79 ± 2.51			
								−2.81	0.005 ^‡^		
Sleep	Exp. (n = 23)	3.70 ± 1.02	5.17 ± 1.67					1.48 ± 1.70			
								−3.23	0.001 ^‡^		
Activity (%)	Exp. (n = 11)	51.18 ± 22.68	81.45 ± 20.90					30.27 ± 29.27			
								−2.81	0.005 ^‡^		

^†^ Mann–Whitney U Test; ^‡^ Wilcoxon Signed Rank Test; T2: 8 weeks; T3: 5 months.

## Data Availability

MDPI Research Data Policies.

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
