# Peer review of "Effectiveness of a Mobile Application for Postpartum Depression Self-Management: Evidence from a Randomised Controlled Trial in South Korea"

_healthcare, 2022, doi:10.3390/healthcare10112185_

Round 1

Reviewer 1 Report (Previous Reviewer 1)

Dear Authors, 

Thank you for the improvements that you made to your paper. You did a nice job considering the suggestions and the explanation for each topic that was unclear to the readers. 

I have no more comments, and I hope you are improving your app to continue helping mothers with postpartum depression. 

Thank you for sharing.

Author Response

Thank you for your valuable comments.

Reviewer 2 Report (Previous Reviewer 2)

This study aimed to examine the effectiveness of a mobile app developed for the self-management of postpartum depression among South Korean mothers. The research topic is indeed interesting and needed. These are my comments and suggestions:

Abstract:

The abstract is incomplete. It should include all the most important information from the manuscript. For instance, outcome measures are missing, and conclusion should be rephrased.

Introduction:

Please give more information regarding this sentence: "As a result of the app usability evaluation, it was confirmed that the quality of the app was superior." How did you confirmed the app superior quality? Do you have any reference?

Methods:

Who developed the app (include manufacturer)? Is there a reference? Give more data on how this app actually works. Include data regarding previous testing of the app. Was the RCT prospectively registered? Give more details regarding the allocation to the groups.

Results:

No comments:

Discussion:

Include your main finding at the beginning of the Discussion and add few sentences why your study is important.

Round 2

Reviewer 2 Report (Previous Reviewer 2)

I am happy with the improvements of the article. However, this trial was not prospectively registered and the overall merit of this study is quite low.

This manuscript is a resubmission of an earlier submission. The following is a list of the peer review reports and author responses from that submission.

Round 1

Reviewer 1 Report

Dear Authors, 

I found your study interesting in the topic that you are focusing on. I have some comments. 

Abstract

Line 25. "The app would be helpful as a part of self-management nursing intervention...." I consider that you could change the last phrase to be more specific about what your results confirmed. The app could be more beneficial in promoting healthy behavior. 

Introduction 

Line 43, I will remove "Advanced" I would suggest only "Countries such US and UK..."

You made clear in your introduction that Korean mothers consider postpartum depression a natural change. However, could you explain more about the increase in the percentage of Korean women suffering from postpartum depression? What are now the conditions that changed or made this problem more visible?

Line 61-62. Please, consider explaining that this is a tool that helps the mothers to figure out their situation. Please consider rearranging the text because cost-effectiveness and easy access would never replace standard psychological treatment. 

Line 111, please correct the parenthesis 

4. Discussion

Line 345, could you please explain better at the discussion the point "shifting common negative thoughts into positive ones"

Line 448, I agree that measuring the app's effect on mothers is difficult. This is why I consider that you should emphasize its use in healthy behaviors, not necessarily in postpartum depression or anxiety. 

It would help if you also consider that depression could be affected by diet. It would be interesting to add to your app the "remained" of decreased free sugars and lipids and increased intake of phytonutrients and flavonoids (mandarins, apples, grapes, persimmons, soybeans). 

Thank you for your effort in your paper. 

Reviewer 2 Report

This study aimed to examine the effectiveness of a mobile app developed for the self-management of postpartum depression among Korean mothers. The research topic is indeed interesting and needed. These are my comments and suggestions:

Abstract:

I suggest to add more quantitative data in the abstract. Abstract should include only data from the investigated outcomes. Some conclusions regarding the update of the app and practice of use are not necessary in the abstract.

Introduction:

Some punctuation marks are missing in the text. I advise not to divide Introduction chapter into subchapters.

Methods:

Please, include design of the study at the start of the Methods chapter. Who developed the app? Is there a reference? Include data regarding previous testing of the app. Was the RCT prospectively registered? Give more details regarding the allocation to the groups. Was your data normally distributed to use RM ANOVA? Why did you use T-test for independent samples and Mann Whitney U test for comparison of pre- and post- measurements? If this was between-group analysis then be more specific.

Results:

Tables should be formatted in a different way (they are outside of the page margins).

Discussion and conclusion:

Nicely written. No comments to add. However, scientific merit is modest at least.

Reviewer 3 Report

-Please proofread the article, especially for punctuation.

- Is this a self-made app that was designed by the study team or is this app widely used? How would the woman who just gave birth to her 3rd child benefit from learning about the role of a mother? Also would women who do not have thoughts the app qualified as negative relate to the app? Sometimes postpartum depression looks like lack of motivation and low energy, not necessarily negative thoughts/behaviors.  Also not all women with PPD are unhappy, so the wording of a happiness diary may again make the usage of the app hard for some women to relate to.

- How were the 14 out of 18 questions about depression knowledge selected? Same for maladaptive beliefs, stress-coping behavior,

- Would a woman suffering from depression join an internet community? For the woman who was overwhelmed and feeling inadequate what would be the rationale for joining a community where everyone else seems to "have it together". Is this a limitation or a population of convenience?

- The bulletin board is a positive feature of the app

- Did women with more adherence or usage of the app have better depression or coping skills relative to women with less adherence and usage? Similarly were there any correlations between depression and coping skills with age? Number of children? Length of time since birth? Birth complications? All of those factors can impact the study results.